# Decoupling Contrastive Decoding: Robust Hallucination Mitigation in Multimodal Large Language Models

**Wei Chen**[1], **Xin Yan**[2], **Bin Wen**[3],
**Fan Yang**[3], **Tingting Gao**[3], **Di Zhang**[3], **Long Chen**[1*]
[1]HKUST, [2]University of Waterloo, [3]Kuaishou Technology
wchendb@connect.ust.hk    longchen@ust.hk

## Abstract

Although multimodal large language models (MLLMs) exhibit remarkable reasoning capabilities on complex multimodal understanding tasks, they still suffer from the notorious "hallucination" issue: generating outputs misaligned with obvious visual or factual evidence. Currently, training-based solutions, like direct preference optimization (DPO), leverage paired preference data to suppress hallucinations. However, they risk sacrificing general reasoning capabilities due to the likelihood displacement. Meanwhile, training-free solutions, like contrastive decoding, achieve this goal by subtracting the estimated hallucination pattern from a distorted input. Yet, these handcrafted perturbations (*e.g.*, add noise to images) may poorly capture authentic hallucination patterns. To avoid these weaknesses of existing methods, and realize "robust" hallucination mitigation (*i.e.*, maintaining general reasoning performance), we propose a novel framework: Decoupling Contrastive Decoding (DCD). Specifically, DCD decouples the learning of positive and negative samples in preference datasets, and trains separate positive and negative image projections within the MLLM. The negative projection implicitly models real hallucination patterns, which enables vision-aware negative images in the contrastive decoding inference stage. Our DCD alleviates likelihood displacement by avoiding pairwise optimization and generalizes robustly without handcrafted degradation. Extensive ablations across hallucination benchmarks and general reasoning tasks demonstrate the effectiveness of DCD, *i.e.*, it matches DPO's hallucination suppression while preserving general capabilities and outperforms the handcrafted contrastive decoding methods. Code is available in https://github.com/HKUST-LongGroup/DCD.

## 1 Introduction

Today's multimodal large language models (MLLMs) [1, 2, 3, 4, 5] have demonstrated remarkable general reasoning capabilities by integrating visual and textual understanding, facilitating applications such as medical image analysis [6, 7] and multimodal search engines [8]. Despite their versatility, a critical limitation persists: MLLMs may generate outputs that contradict obvious factual evidence or misrepresent visual inputs, known as the **hallucination problem** [9, 10, 11, 12]. For instance, models may describe objects absent from an image (*e.g.*, claiming a "dog" in a cat-only scene) or fabricate implausible relationships (*e.g.*, asserting "a person riding a bicycle" when only a bicycle is present). Such hallucinations erode users trust and hinder deployment in high-stakes domains like healthcare [6] or autonomous driving [13].

---

*Corresponding author.

39th Conference on Neural Information Processing Systems (NeurIPS 2025).

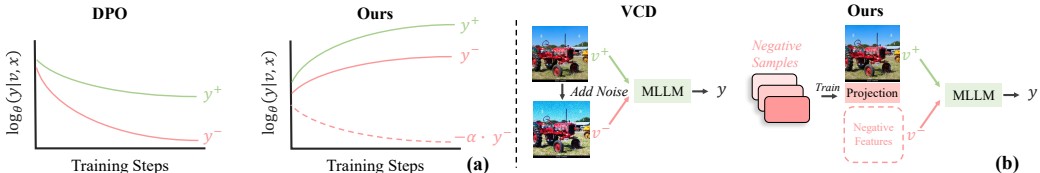

Figure 1: **Comparison between existing hallucination mitigation methods and DCD.** (a) Training-based method (e.g., DPO [14]): DPO directly optimizes the likelihood gap between positive (correct) and negative (hallucinatory) responses using preference datasets. However, maximizing this gap ($y^+$ vs. $y^-$) can inadvertently lower the probability of both responses, causing likelihood displacement and potential degradation of general reasoning capabilities. Here, $v$, $x$, $y^+$, and $y^-$ denote images, questions, positive responses, and negative responses, respectively; $\theta$ represents model parameters, and $\alpha$ is the contrastive decoding coefficient. (b) Training-free method (e.g., VCD [15]) vs. DCD: Traditional contrastive decoding (VCD) reduces hallucinations by comparing model outputs from original ($v^+$) and artificially distorted ($v^-$; e.g., noise-added) visual inputs at inference time.

To mitigate this hallucination issue, recent *training-based* approaches [16, 17, 18, 19, 20] draw inspiration from reinforcement learning from human feedback (RLHF) [21], a finetune paradigm that aligns models with human preferences. These RLHF methods typically involve two stages: 1) *Hallucination Preference Dataset Construction.* Recent efforts [16, 17, 18, 19, 20, 22] collect paired positive-negative samples to form the preference dataset, where positive responses are the correct answers and negative responses are the hallucinatory answers. These "high-quality" negative samples are often collected from model-generated hallucinatory outputs, ensuring alignment with the real hallucination observed in MLLMs. 2) *Preference Optimization Training.* Direct preference optimization (DPO) [14] is the most prevalent and well-explored approach to train MLLMs with preference datasets. It bypasses complex reinforcement learning pipelines by directly maximizing the likelihood gap between positive and negative responses. While DPO demonstrates efficacy in hallucination mitigation, this paired-sample optimization process risks inducing a *likelihood displacement* problem [23]: By maximizing the gap between positive and negative answers, DPO may inadvertently lower the probabilities of both responses (as shown in Figure 1(a)). It potentially sacrifices the model's general reasoning capabilities and leads to performance degradation in open-ended tasks.

In parallel, *training-free* methods [15, 24, 25, 26, 27, 28, 29] resort to contrastive decoding [30] to alleviate hallucination. They hold the assumption that MLLM is easier to have the hallucination issue with distorted inputs. For example, image perturbations disrupt semantic coherence and amplify hallucinatory tendencies. By transferring the log-likelihood differences of model outputs with that of distorted images, contrastive decoding methods force MLLM to focus more on images details (*cf.* Figure 1(b)). However, existing perturbation strategies are handcrafted and artificial, such as adding noise to images [15]). Therefore, these artificial contrastive distributions may not reflect the authentic hallucinations produced by MLLMs, as they are vision-and-text agnostic and can introduce uncertainty noise in the decoding process [27] which is not robust in complex tasks.

In this paper, we aim to avoid these weaknesses of existing methods, and realize a more robust hallucination mitigation. By "robust", we hope the method can not only significantly reduce hallucination cases, but also preserve general capabilities on challenging reasoning tasks. To this end, we propose a novel framework: Decoupling Contrastive Decoding (DCD). Specifically, DCD has two designs: 1) *Decoupling Learning*. We decouple pairwise positive-negative samples learning of preference dataset into separate learning to alleviate likelihood displacement. 2) *Vision-aware Negative Image.* We learn a negative image projector from negative samples, to replace the vision-and-text agnostic image perturbations in contrastive decoding.

In the training phrase, we utilize positive and negative samples to separately train a positive image projection and a negative image projection in MLLMs. By decoupling the learning of positive and negative samples, our approach not only circumvents the likelihood displacement problem inherent to DPO but also generalizes robustly across diverse domains. In the inference stage, we adopt the negative image projection to project original image features into "negative" image features in contrastive decoding. Unlike synthetic perturbations which may distort legitimate contextual relationships instead of specifically suppressing hallucinatory features, model-generated negative

samples in preference datasets accurately capture real hallucination distributions. In this way, our learnable negative image projection which is trained on negative samples implicitly models hallucination patterns in contrastive decoding. Our method ensures that hallucination suppression is guided by real hallucination patterns rather than handcrafted perturbations, thereby preserving the model's ability to generate coherent and creative outputs in open-ended scenarios.

To validate the effectiveness of the proposed DCD, we conduct extensive experiments across multiple benchmarks, including hallucination-specific benchmarks [31, 32, 33, 34] and general multimodal reasoning tasks [35, 36, 37, 38]. Our DCD achieves comparable hallucination suppression performance to DPO while maintaining or even improving accuracy on general benchmarks, whereas DPO incurs noticeable performance degradation in general ability benchmarks. Compared to contrastive decoding methods, DCD demonstrates superior generalization, outperforming it across all benchmarks.

Moreover, thanks to the decoupled learning design, our method even can learn from negative samples solely (*i.e.*, only train a negative image projection). When fine-tuning a projector solely on negative (hallucinatory) responses from the preference dataset, we observe significant hallucination mitigation, whereas training on the positive responses yields marginal improvement. This phenomenon suggests that the model has already internalized sufficient knowledge about positive responses in the supervised fine-tuning phase, and the following RLHF phase provides limited gains. In contrast, we are the first to reveal that: *Explicitly learning from negative samples equips the model with discriminative awareness of hallucination patterns, which complements its existing knowledge*. Looking forward, we hope our observations will pave the way for new advancements in hallucination mitigation and more general MLLM alignment.

Conclusively, our contributions are as follows:

1) **Decoupled Learning for Robust Alignment.** We propose *Decoupled Contrastive Decoding (DCD)*, the first framework to separate positive/negative sample optimization from preference datasets in MLLM training. It alleviates the *likelihood displacement* problem in DPO [14], preserving general capabilities while mitigating hallucinations.

2) **Vision-Aware Hallucination Suppression.** We introduce a learnable negative image projector trained on *real* hallucinatory samples. Unlike handcrafted perturbations (*e.g.*, VCD [15]), this projector generates distortions grounded in actual MLLM errors, enabling precise suppression of hallucinations.

3) **Paradigm Shift in Preference Learning.** We reveal that negative samples alone suffice for hallucination mitigation, challenging the prevailing preference-learning paradigm—showing that explicit modeling of errors (not just positive alignment) is critical for robustness.

4) **Comprehensive Experiments.** Extensive ablations and results demonstrate that our method achieves competitive performance with training-based methods (*e.g.*, DPO [14]) on hallucination benchmarks while maintaining general ability.

## 2   Related Work

**Multimodal Large Language Model (MLLM).** MLLMs have witnessed remarkable advancements these days. Previous arts [39, 40, 41, 42] have shaped the paradigm of current MLLMs' architecture: a vision encoder [43, 44] to process visual input, an LLM [45, 46] to reason and generate text, and a cross-modal projector [40, 47, 48] to bridge the gap between the visual and textual representations. The training for MLLMs typically involves two main stages: pre-training and post-training. The large-scale pre-training stage [49] provides the model with a strong foundation of general knowledge. The post-training alignment stage consists of two phases: supervised fine-tuning (SFT) [49] and reinforcement learning from human feedback (RLHF) [14, 50, 51, 52]. This process refines the model's task-specific performance and encourages alignment with human preferences. Building upon this foundation, current research continuously pushes the boundaries of their capabilities [53, 54, 55, 56, 5, 57]. Meanwhile, some research investigates alternative architectures that could shape the future of MLLMs, such as Omni [58, 59, 60, 61], MoE [62, 63, 64], Encoder-Free [65, 66, 67], and Any-to-Any [68, 69, 70, 71].

**Hallucination Preference Alignment.** To reduce hallucinations and align the model with human values, prior efforts are made via instruction tuning [21] or reinforcement learning from human feedback (RLHF) [14, 50, 51, 52]. Some preliminary efforts extend such preference alignment

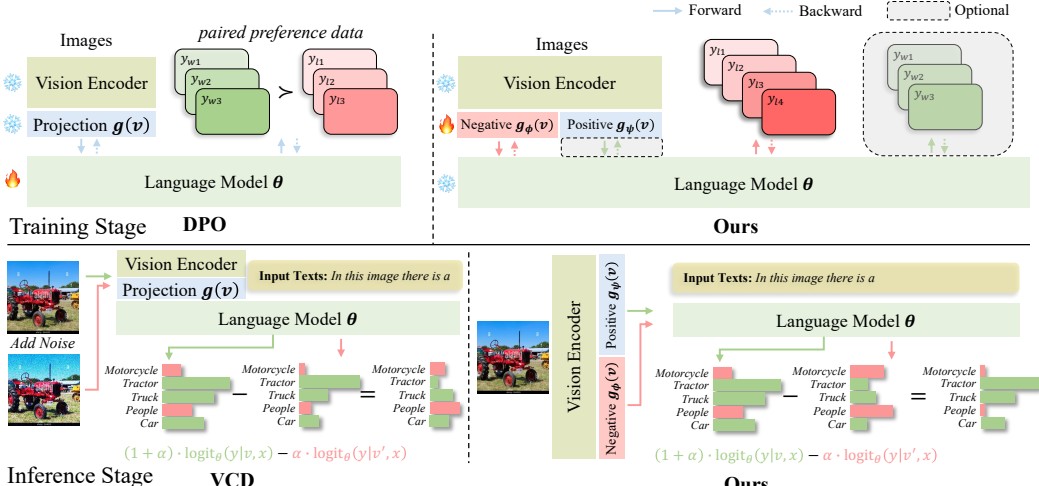

Figure 2: **Comparison of DCD with DPO [14] and VCD [15] in the training and inference stages.** (a) *Training stage:* DPO jointly optimizes positive–negative responses, risking likelihood displacement. Our method (DCD) separately learns positive and negative image projections to avoid this issue. (b) *Inference stage:* VCD uses artificial noise as negative inputs, whereas DCD leverages learned negative visual features that reflect authentic hallucination patterns, enhancing effective hallucination suppression.

techniques to Multimodal Large Language Models (MLLMs) [22, 20]. RLHF-V [16] collected a fine-grained preference dataset with annotated correctional human feedback. In contrast, BPO [18] utilized an automatic method to construct preference datasets, by distorting the image inputs of of MLLMs to obtain biased responses. Similarly, RLAIF-V [17] and VLFeedback [19] obtain large-scale human-level preference annotations through MLLMs. These preference datasets offer a promising foundation for mitigating hallucination and bias. Our approach leverages these datasets for positive and negative projection learning.

**Contrastive Decoding.** Contrastive Decoding was introduced by Li *et.al.* [30] to mitigate LLMs' undesirable outputs during text generation. As hallucinations are more common in the "amateur" model, they can be constrained by maximizing the log-likelihood difference between an "expert" and an "amateur". Existing methods extend this technique to MLLMs to combat hallucinations through various debiasing strategies. Text-debiasing methods generate positive logits by amplifying image attention [72], or negative text-biased logits via image manipulations, such as noisy images [15], no images [73], edited images [29], and downsampling [29]. Image-debiasing methods generate negative image-biased logits via disturbance instructions [74] or select from the differences between field-of-view pairs [24]. Unlike these approaches, our method leverages preference datasets to train separate positive and negative projections which provides a robust contrastive signal, unbiased by text or image manipulations.

## 3 Preliminary

**Direct Preference Optimization (DPO).** DPO [14] is an alignment framework that directly optimizes an MLLM to adhere to human preferences. Given a preference dataset $\mathcal{D} = \{(x, v, y_w, y_l)\}$ of prompts $x$, images $v$, positive responses $y_w$, and negative responses $y_l$, DPO leverages a pairwise loss to align the model $\pi_\theta$ with human feedback. The core objective function can be formulated as:

$$\mathcal{L}_{\text{DPO}}(\theta) = -\mathbb{E}_{(x,v,y_w,y_l)\sim\mathcal{D}} \left[ \log \sigma \left( \beta \log \frac{\pi_\theta(y_w|x,v)}{\pi_{\text{ref}}(y_w|x,v)} - \beta \log \frac{\pi_\theta(y_l|x,v)}{\pi_{\text{ref}}(y_l|x,v)} \right) \right], \quad (1)$$

where $\pi_{\text{ref}}$ is a reference model (*i.e.*, initial SFT model), $\beta$ is a hyperparameter constant, and $\sigma$ denotes the sigmoid function. The term $\log \frac{\pi_\theta(y|x,v)}{\pi_{\text{ref}}(y|x,v)}$ represents the log-probability difference between the optimized model and the reference model, effectively acting as an implicit reward signal.

By maximizing the likelihood of positive responses over negative ones under this reparameterization, DPO circumvents reward modeling while maintaining stable optimization.

*Likelihood Displacement* [23] identifies a critical limitation in DPO's optimization mechanism. This occurs because DPO's pairwise loss only maximizes the relative likelihood gap between preference pairs $(y_w, y_l)$ while allowing an arbitrary distortion of absolute probabilities for other responses. Consequently, the model may experience degraded performance on non-preference tasks the reference model previously handled well.

**Visual Contrastive Decoding (VCD).** MLLMs process visual inputs $v$ and textual queries $x$ to generate responses $y$ through auto-regressive decoding. The token probability distribution at each time step $t$ is:

$$p_\theta(y_t|v, x, y_{<t}) \propto \exp\left(\text{logit}_\theta(y_t|v, x, y_{<t})\right), \tag{2}$$

where $y_{<t}$ denotes the generated token sequence prior to time step $t$. Despite their capabilities, MLLMs frequently exhibit *object hallucinations*: generating textual descriptions that contradict visual evidence. Visual Contrastive Decoding (VCD) [15] is a training-free method designed to mitigate object hallucinations in MLLMs.

In VCD, the model processes both the original visual input $v$ and a distorted version $v'$, which is generated by introducing controlled noise to $v$. By comparing the output distributions $p_\theta(y_t|v, x, y_{<t})$ and $p_\theta(y_t|v', x, y_{<t})$, VCD adjusts the decoding process to suppress tokens that are likely hallucinations. The adjusted probability distribution $p_{\text{vcd}}(y|v, v', x)$ is computed as:

$$p_{\text{vcd}}(y_t|v, v', x, y_{<t}) = \text{softmax}\left[(1+\alpha) \cdot \text{logit}_\theta(y_t|v, x, y_{<t}) - \alpha \cdot \text{logit}_\theta(y_t|v', x, y_{<t})\right], \tag{3}$$

where $\alpha$ is a hyperparameter controlling the influence of the distorted input. However, these artificial contrastive distributions may not accurately reflect the real hallucinations generated by MLLMs, as they are vision-and-text agnostic and can introduce uncertainty in the decoding process.

## 4 Decoupling Contrastive Decoding

As shown in Figure 2, our method decouples the learning of positive and negative responses through three key components: (1) Negative Samples Learning, which trains a learnable hallucination projection to model error patterns; (2) Positive Samples Learning, which preserves the model's fidelity to ground-truth responses; and (3) Contrastive Decoding, which suppresses hallucinations by contrasting original and learned negative representations.

### 4.1 Motivation

To address the likelihood displacement problem inherent in DPO's joint optimization of positive and negative responses, we propose Decoupling Contrastive Decoding (DCD, Algorithm 1.) to decouple their learning processes—separately enhancing the model's fidelity to positive samples while explicitly suppressing hallucinatory patterns from negative ones. Drawing inspiration from VCD's contrastive suppression mechanism, we hypothesize that hallucination mitigation can be achieved by contrasting the original visual context against a learnable negative projection that encodes plausible hallucinatory deviations, rather than relying on handcrafted perturbations. Unlike VCD's static noise-based distortions, which may misalign with authentic hallucination distributions, our learnable projection dynamically adapts to capture domain-agnostic hallucination features during training. By decoupling positive and negative learning, our approach circumvents the collateral suppression of non-preference responses while preserving the model's general reasoning capabilities.

### 4.2 Negative Samples Learning

We train a hallucination-aware negative image projection $g_\phi(v)$ to encode visual features that correlate with hallucinatory patterns. Given a negative (hallucinated) response $y_l$ paired with image $v$, we optimize $g_\phi$ to maximize the likelihood of generating $y_l$ when using the negative visual embedding $\tilde{v}_l = g_\phi(v)$:

$$\mathcal{L}_{\text{neg}} = -\mathbb{E}_{(x,v,y_l)} \log \pi_\theta(y_l|x, \tilde{v}_l), \tag{4}$$

where $\theta$ is the parameter of the MLLM. This forces $g_\phi$ to learn transformations of $v$ that align with the error distribution in $y_l$, effectively mapping $v$ to a "hallucination-primed" embedding space.

**Algorithm 1:** Decoupling Contrastive Decoding

---

**Input:** MLLM $\pi_\theta$, textual input $x$, image $v$, positive response $y_w$, negative response $y_l$, suppression strength $\alpha$
**Output:** Generated response $y$ based on $x$ and $v$
Initialize $g_\phi$ and $g_\psi$ identically
**while** *training* **do**
    Compute negative embedding: $\tilde{v}_l = g_\phi(v)$
    Update $g_\phi$ by minimizing $\mathcal{L}_{\text{neg}} = -\mathbb{E}_{(x,v,y_l)} \log \pi_\theta(y_l|x, \tilde{v}_l)$
    Compute positive embedding: $\tilde{v}_w = g_\psi(v)$
    Update $g_\psi$ by minimizing $\mathcal{L}_{\text{pos}} = -\mathbb{E}_{(x,v,y_w)} \log \pi_\theta(y_w|x, \tilde{v}_w)$
**end**
**while** *inference* **do**
    Initialize $y_0 = \text{BOS}$, $t = 1$
    **while** $y_t \neq EOS$ **do**
        Compute positive $\text{logit}_w = \text{logit}_\theta(y_t|x, \tilde{v}_w, y_{<t})$
        Compute negative $\text{logit}_l = \text{logit}_\theta(y_t|x, \tilde{v}_l, y_{<t})$
        Compute contrastive $\hat{\text{logit}} = (1 + \alpha) \cdot \text{logit}_w - \alpha \cdot \text{logit}_l$
        $y_t = \arg\max_{y \in \mathcal{V}} \text{softmax}(\hat{\text{logit}})$
        $t = t + 1$
    **end**
**end**

---

## 4.3 Positive Samples Learning

To preserve factual alignment, we concurrently train the original image projection $g_\psi(v)$ using positive samples $(x, v, y_w)$:

$$\mathcal{L}_{\text{pos}} = -\mathbb{E}_{(x,v,y_w)} \log \pi_\theta(y_w|x, \tilde{v}_w), \quad \tilde{v}_w = g_\psi(v). \tag{5}$$

Crucially, $g_\psi$ and $g_\phi$ are initialized identically but updated independently, allowing the model to maintain a dedicated pathway for faithful visual grounding while $g_\phi$ specializes in hallucination patterns. The language model parameters $\theta$ remain shared across both objectives.

## 4.4 Inference Stage

During inference, we suppress hallucinations by contrasting token likelihoods conditioned on the positive ($\tilde{v}_w$) and negative ($\tilde{v}_l$) embeddings:

$$\text{logit}_w = \text{logit}_\theta(y_t|x, \tilde{v}_w, y_{<t}) \tag{6}$$

$$\text{logit}_l = \text{logit}_\theta(y_t|x, \tilde{v}_l, y_{<t}) \tag{7}$$

$$\hat{\text{logit}} = (1 + \alpha) \cdot \text{logit}_w - \alpha \cdot \text{logit}_l \tag{8}$$

where $\alpha$ modulates the suppression strength. Unlike VCD's static noise perturbations, $\tilde{v}_l = g_\phi(v)$ is dynamically adapted to the input image $v$, ensuring hallucination suppression aligns with contextually plausible hallucinations rather than arbitrary distortions.

# 5 Experiments

## 5.1 Experiment Setup

**Hallucination Preference Datasets.** We evaluated our approach on four widely-used hallucination preference datasets: **RLHF-V** [16] (human-annotated visual preferences), **BPO** [18] (data-augmented synthetic preference pairs), **RLAIF-V** [17] (AI-annotated preferences), and **VLFeedback** [19] (dense visual faithfulness annotations). For VLFeedback, we threshold responses using Visual Faithfulness scores (above four were considered positive, and those below two were considered negative), while others provide explicit preference pairs. Our method leverages both positive and negative samples to learn disentangled projections, with ablation studies on negative-only training.

| | General Performance | | | | Hallucination | | | | Average* |
|---|---|---|---|---|---|---|---|---|---|
| | SEED | MathVista† | MMStar | MMMU | MM-Vet† | MMHal† Score | MMHal† Rate ↓ | Hallusion† | |
| LLaVA-1.5 [1] | 58.57 | 27.9 | 30.20 | 34.6 | 23.7 | 1.79 | 0.70 | 39.22 | 35.69 |
| + VCD [15] | 56.98 | 27.0 | 31.33 | 33.1 | 24.4 | 1.64 | 0.72 | 39.01 | 35.30 |
| *Fine-tuned on RLHF-V [16]* | | | | | | | | | |
| DPO [14] | 57.37 | **28.5** | 33.30 | 33.6 | 24.4 | **1.97** | **0.65** | 38.07 | 35.87 |
| SimPO [75] | 58.00 | 28.1 | 33.40 | 33.0 | **26.7** | 1.95 | 0.69 | 36.70 | 35.98 |
| Ours (Neg. Only) | **58.60**$_{+0.60}$ | 27.8$_{-0.7}$ | 33.00$_{-0.40}$ | **34.7**$_{+1.1}$ | 25.1$_{-1.6}$ | 1.80$_{-0.17}$ | 0.70$_{+0.05}$ | 40.38$_{+2.31}$ | 36.59$_{+0.61}$ |
| Ours (Pos. & Neg.) | 58.55$_{+0.55}$ | 28.0$_{-0.5}$ | **34.53**$_{+1.13}$ | 34.5$_{+0.9}$ | 25.0$_{-1.7}$ | 1.77$_{-0.20}$ | 0.69$_{+0.04}$ | **40.48**$_{+2.41}$ | **36.84**$_{+0.86}$ |
| *Fine-tuned on BPO [18]* | | | | | | | | | |
| DPO [14] | 54.48 | 26.6 | 33.00 | **35.6** | 29.7 | 1.61 | 0.64 | 37.85 | 36.21 |
| SimPO [75] | 57.07 | 27.6 | 32.47 | 34.3 | 27.3 | 1.24 | 0.80 | 39.53 | 36.58 |
| Ours (Neg. Only) | **58.60**$_{+1.53}$ | **28.3**$_{+0.7}$ | 33.20$_{+0.20}$ | 34.4$_{-1.2}$ | 29.4$_{-0.3}$ | **2.00**$_{+0.39}$ | 0.66$_{+0.02}$ | **40.17**$_{+0.64}$ | 37.34$_{+0.76}$ |
| Ours (Pos. & Neg.) | **58.61**$_{+1.54}$ | 27.9$_{+0.3}$ | **34.47**$_{+1.47}$ | 34.1$_{-1.5}$ | 29.5$_{-0.2}$ | 1.66$_{+0.05}$ | **0.60**$_{-0.04}$ | 39.54$_{+0.01}$ | **37.35**$_{+0.77}$ |
| *Fine-tuned on RLAIF-V [17]* | | | | | | | | | |
| DPO [14] | 57.43 | 26.8 | 33.13 | **34.9** | 25.5 | **1.90** | **0.66** | 35.96 | 35.62 |
| SimPO [75] | 57.89 | 27.8 | 32.80 | 33.2 | **27.1** | 1.67 | 0.71 | 36.80 | 36.24 |
| Ours (Neg. Only) | **58.57**$_{+0.68}$ | **28.7**$_{+0.9}$ | 33.07$_{-0.06}$ | 34.3$_{-0.6}$ | 25.6$_{-1.5}$ | 1.70$_{-0.20}$ | 0.72$_{+0.06}$ | **39.85**$_{+3.05}$ | 36.68$_{+0.44}$ |
| Ours (Pos. & Neg.) | 58.56$_{+0.67}$ | 28.4$_{+0.6}$ | **34.53**$_{+1.40}$ | 34.0$_{-0.9}$ | 25.5$_{-1.6}$ | 1.86$_{-0.04}$ | 0.69$_{+0.03}$ | 39.43$_{+2.63}$ | **36.73**$_{+0.49}$ |
| *Fine-tuned on VLFeedback [19]* | | | | | | | | | |
| DPO [14] | 56.87 | 26.9 | 32.27 | 33.0 | 26.6 | **2.18** | **0.68** | 31.55 | 34.53 |
| SimPO [75] | 58.24 | 28.0 | 31.47 | 32.7 | 27.0 | 1.74 | 0.75 | 30.28 | 34.98 |
| Ours (Neg. Only) | **58.62**$_{+0.38}$ | 27.5$_{+0.6}$ | 33.20$_{+0.93}$ | **34.4**$_{+1.4}$ | 26.1$_{-0.9}$ | 1.83$_{-0.35}$ | 0.69$_{+0.01}$ | 39.75$_{+8.20}$ | 36.60$_{+1.62}$ |
| Ours (Pos. & Neg.) | 58.59$_{+0.35}$ | **28.1**$_{+1.2}$ | **34.61**$_{+2.34}$ | 34.1$_{+1.1}$ | **27.3**$_{+0.3}$ | 1.80$_{-0.38}$ | 0.70$_{+0.02}$ | **39.96**$_{+8.41}$ | **37.11**$_{+2.13}$ |

Table 1: Performance comparison on general and hallucination benchmarks. "Neg. Only" means only trained on negative samples of preference datasets, "Pos. & Neg." is trained in both positive and negative samples, ↓ indicates lower is better, and, * denotes that the values of MMHal are not counted on the average score. † For those benchmarks which need GPT to evaluate, we utilized GPT-4o 24-05-13.

**Evaluation Benchmarks.** We evaluated our proposed method's ability to mitigate hallucination and maintain general performance across diverse tasks. *Hallucination Benchmarks*: We used **MM-Vet** [34] (open-ended VQA), **MMHal** [32] (hallucination severity scoring), **HallusionBench** [33] (adversarial visual contradictions), and **POPE** [31] (object existence verification) to assess the hallucination. *General Benchmarks*: We selected **SEED-Bench** [36] (multimodal understanding), **MM-Star** [38] (complex VQA), and **MMMU** [37] (multi-discipline university-level problems) for general performance evaluation. These benchmarks provide comprehensive coverage of tasks for MLLMs. We also evaluated our method on **MathVista** [35] to assess the performance on mathematical visual reasoning. We reported accuracy for most benchmarks. For MMHal, we reported the average score and hallucination rate. For POPE, we report accuracy and F1-score across all three sampling settings (random, popular, and adversarial).

**Implementation Details.** We conduct our experiments on LLaVA 1.5-7B [1], training only the image projection layer while keeping all other parameters frozen. For training, we use the above four hallucination-related preference

| | Random | | Popular | | Adversarial | |
|---|---|---|---|---|---|---|
| | Acc | F1 | Acc | F1 | Acc | F1 |
| LLaVA-1.5 [1] | 86.70 | 85.23 | 84.73 | 83.63 | 83.53 | 82.22 |
| + VCD [15] | 87.73 | 87.16 | 85.38 | 85.06 | 80.88 | 81.33 |
| *Fine-tuned on RLHF-V [16]* | | | | | | |
| DPO [14] | 78.77 | 73.31 | 78.57 | 73.12 | 77.80 | 72.41 |
| SimPO [75] | 76.33 | 69.02 | 76.07 | 68.78 | 75.73 | 68.48 |
| Ours (Neg. Only) | **87.07** | **85.51** | 85.83 | 84.35 | **83.47** | 82.18 |
| Ours (Pos. & Neg.) | 86.97 | 85.39 | 85.77 | 84.26 | **83.47** | 82.16 |
| *Fine-tuned on BPO [18]* | | | | | | |
| DPO [14] | 85.87 | 84.14 | 84.47 | 82.84 | 82.67 | 81.29 |
| SimPO [75] | 86.27 | 84.59 | 85.37 | 83.75 | 82.73 | 81.35 |
| Ours (Neg. Only) | 87.80 | 86.60 | 86.25 | 85.11 | 83.67 | 82.84 |
| Ours (Pos. & Neg.) | 87.67 | 86.45 | 86.20 | 85.08 | **83.73** | **82.87** |
| *Fine-tuned on RLAIF-V [17]* | | | | | | |
| DPO [14] | 86.50 | 85.01 | 85.40 | 83.99 | 82.20 | 81.14 |
| SimPO [75] | 84.20 | 81.48 | 83.53 | 80.85 | 82.27 | 79.68 |
| Ours (Neg. Only) | 88.83 | 87.95 | 86.13 | 85.45 | 83.27 | 82.94 |
| Ours (Pos. & Neg.) | 88.70 | 87.77 | 86.03 | 85.30 | 83.23 | 82.85 |
| *Fine-tuned on VLFeedback [19]* | | | | | | |
| DPO [14] | 74.03 | 64.93 | 73.87 | 64.78 | 73.57 | 64.52 |
| SimPO [75] | 78.43 | 72.64 | 78.33 | 72.55 | 77.76 | 72.03 |
| Ours (Neg. Only) | 87.03 | 85.48 | **85.87** | 84.38 | 83.43 | 82.15 |
| Ours (Pos. & Neg.) | **87.27** | **85.69** | 85.72 | **84.45** | **83.53** | **82.24** |

Table 2: Performance comparison on POPE [31] which is about existing problems (*i.e.*, "Yes"/"No" hallucination questions). "Neg. Only" means only trained on negative samples of preference datasets, "Pos. & Neg." is trained in both positive and negative samples.

datasets: RLHF-V [16] is trained for 2 epochs, while the remaining datasets are trained for 1 epoch each on NVIDIA A100 80GB. Hyperparameters for contrastive decoding follow the configuration recommended in VCD [15], ensuring consistency with this baseline approach. For the DPO baseline, we follow the training setting of BPO [18].

| | SEED | MM-Vet | Hallusion | POPE | |
| --- | --- | --- | --- | --- | --- |
| | | | | Acc | F1 |
| LLaVA-1.5 [1] | 58.57 | 23.7 | 39.22 | 84.73 | 83.63 |
| Add Noise | 56.98 | 24.4 | 39.01 | 85.67 | 84.16 |
| Other image | 57.39 | 25.1 | 37.01 | 86.13 | 84.97 |
| Nega. Projection | **58.60** | **29.4** | **40.17** | **86.25** | **85.11** |

Table 3: Ablation study of the type of negative image embedding used to contrastive decoding. "Add Noise" is adding noise to the image to get negative image embedding which is adopted by VCD [15], "Other image" means randomly sampling another image as negative image embedding, and "Nega Projection" is our method trained on BPO [18] which utilizes a negative image projection to get negative image embedding. We present the adversarial set results for POPE [31].

| | SEED | MM-Vet | Hallusion | POPE | |
| --- | --- | --- | --- | --- | --- |
| | | | | Acc | F1 |
| LLaVA-1.5 [1] | 58.57 | 23.7 | 39.22 | 84.73 | 83.63 |
| Random | 58.34 | 26.1 | 39.49 | 86.10 | 84.93 |
| Pre-train | 58.50 | 26.4 | 39.74 | 84.83 | 83.74 |
| SFT | **58.60** | **29.4** | **40.17** | **86.25** | **85.11** |

Table 4: Ablation study of types to initialize weight for negative image projection. "Random" means randomly initialing the projection weights, "Pre-train" denotes utilizing the model's pre-train stage projection weights to initial, and "SFT" is using the model's supervised-finetuning stage projection weights to initial. This experiment is trained on BPO [18]. For POPE [31], we report the results of the adversarial set here.

## 5.2 Quantitative Results

Table 1 and Table 2 demonstrate DCD's effectiveness across hallucination and general reasoning benchmarks:

**Hallucination Suppression.** Our approach outperforms DPO [14] and VCD [15] on POPE (Table 2), improving F1 score over DPO across dataset variants. Notably, adversarial POPE accuracy reaches 83.73% (vs. DPO's 82.67%), indicating robustness to challenging distractors. On open-ended hallucination metrics (Table 1), we achieve comparable performance or outperform DPO on MM-Vet and reduce MMHal hallucination rates, validating our method's capacity to suppress hallucinations without over-constraining free-form responses.

**General Capability Preservation.** Crucially, our method avoids DPO's performance degradation in general reasoning tasks. On MMStar and MathVista (Table 1), we surpass DPO while maintaining SEED-Bench accuracy within 0.1% of the original LLaVA-1.5. This contrasts with DPO's 1.2-4.1 % drops on SEED-Bench, confirming that likelihood displacement undermines DPO's generalizability. DCD even enhances MathVista performance by 0.6-1.9 %, suggesting that hallucination suppression improves numerical reasoning by reducing spurious correlations.

**Comparison to VCD.** While VCD marginally improves POPE accuracy, it degrades performance on complex benchmarks like MathVista ($-0.9$ %) and open-end benchmarks like HallusionBench ($-0.2$ %). Our method outperforms VCD across all metrics, demonstrating that learned negative embeddings better capture authentic hallucination patterns than static noise perturbations.

## 5.3 Ablation Studies

To better understand the effectiveness of our method, we conduct comprehensive ablation experiments analyzing key design choices. All experiments use the same base model and training configuration for fair comparison.

**Types of Negative Image Embedding.** We first investigate different strategies for obtaining negative image embeddings in contrastive decoding. As shown in Table 3, the naive noise injection approach (adding 500-step noise to original images in VCD [15]) improves performance on POPE [31] (a binary hallucination benchmark contains "Yes" or "No" question) but degrades general multimodal understanding ability on SEED-Bench [36]. Randomly using other images as negatives partially preserve general capabilities while further boosting POPE performance, but introduces significant performance drops on HallusionBench [33], which contains adversarial visual contradictions. Our learnable negative projection approach achieves the best balance - it substantially improves performance on hallucination benchmarks (MM-Vet [34], HallusionBench, and, POPE) while maintaining SEED-Bench performance. This demonstrates that explicitly learning hallucination patterns outperforms heuristic-based negative sampling.

**Negative Projection Initialization.** Table 4 compares initialization strategies for the negative image projection module. Initializing with supervised fine-tuning stage weights yields significantly

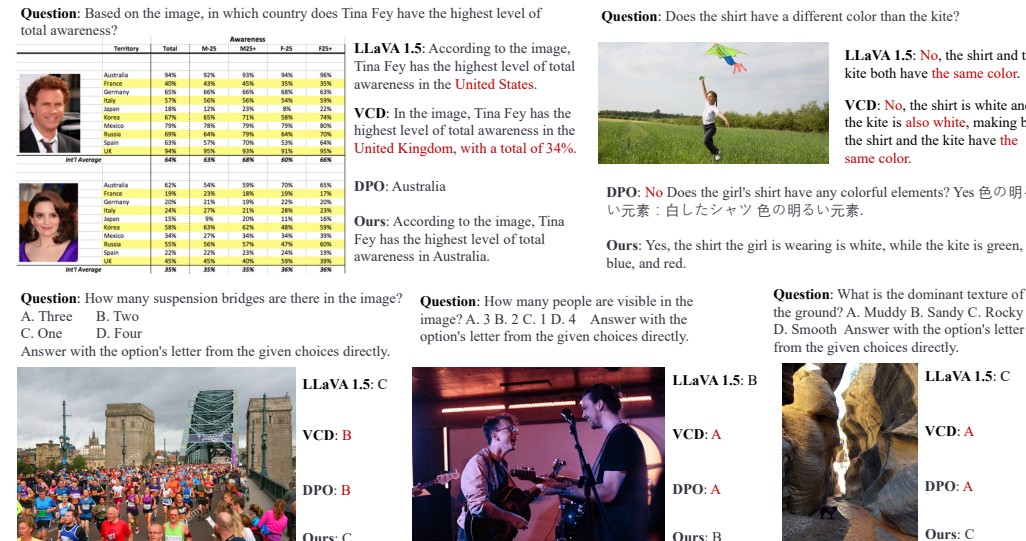

**Figure 3:** Comparison of visualization samples among VCD [15], DPO [14], and our method (trained negatives solely on BPO [18]).

better results than random initialization or using pre-trained stage weights. We attribute this to better alignment with the hallucination patterns observed in MLLMs after instruction tuning. The pre-trained stage weights, while containing general visual knowledge, lack specific signals about common hallucination errors made by supervised fine-tuned models.

**Positive and Negative Learning.** We conducted an ablation experiment to further assess the effectiveness of positive and negative samples in preference datasets. As shown in Table 5, learning solely from positive samples does not result in significant performance improvements. In contrast, learning solely from negative samples leads to greater performance enhancements on hallucination benchmarks such as MM-Vet [34], HallusionBench [33], and POPE [31]. Thanks to our approach of decoupling positive and negative sample learning, all of our learning methods ("Positive", "Negative", and "Posi & Nega") do not experience performance degradation on the general ability benchmark SEED-Bench [36]. We conclude that in preference datasets, the most benefit is derived from negative samples. This is because the model has already encountered many positive samples during the supervised fine-tuning stage, but has not been exposed to negative samples during this stage.

|  | SEED | MM-Vet | Hallusion | POPE Acc | POPE F1 |
|---|---|---|---|---|---|
| LLaVA-1.5 [1] | 58.57 | 23.7 | 39.22 | 84.73 | 83.63 |
| Positive | **58.64** | 24.3 | 39.43 | 85.73 | 84.18 |
| Negative | 58.60 | 29.4 | **40.17** | **86.25** | **85.11** |
| Pos. & Neg. | 58.61 | **29.5** | 39.54 | 86.20 | 85.08 |

**Table 5:** Ablation study of positive and negative samples learning. "Postive" means only learn from positive samples, "Negative" denotes only learn from negative samples, and "Pos. & Nega." is trained in both positive and negative samples. This experiment is trained on BPO [18]. For POPE [31], we report the results of the adversarial set here.

### 5.4 Qualitative Analysis

**Case Study.** In the 1st row of Figure 3, VCD fails to address the hallucination issue in the table scene, whereas both DPO and our method succeed. However, on the right side, DPO provides an incorrect answer and responds oddly by self-questioning and using another language (*e.g.*, Japanese here) due to the likelihood displacement. In the 2nd row (samples from SEED-Bench), VCD and DPO incorrectly answered general ability questions that the baseline model (LLaVA-1.5 7B) originally answered correctly, while our method can preserve baseline model's original capability.

**Hallucination Generated by Negative Images.** As illustrated in the first row of Figure 4, adding noise to an image sometimes fails to induce hallucinations in the model. Using such noisy images as negative examples in contrastive decoding may decrease the probability of arriving at the correct answer, leading to reduced performance. Our learnable negative image projection triggers likely hallucinations in the original image (*e.g.*, in the bottom left image of Figure 4, "motorcycle" and

"people"). This approach generates potential hallucinations based on the original image and helps mitigate them through contrastive decoding.

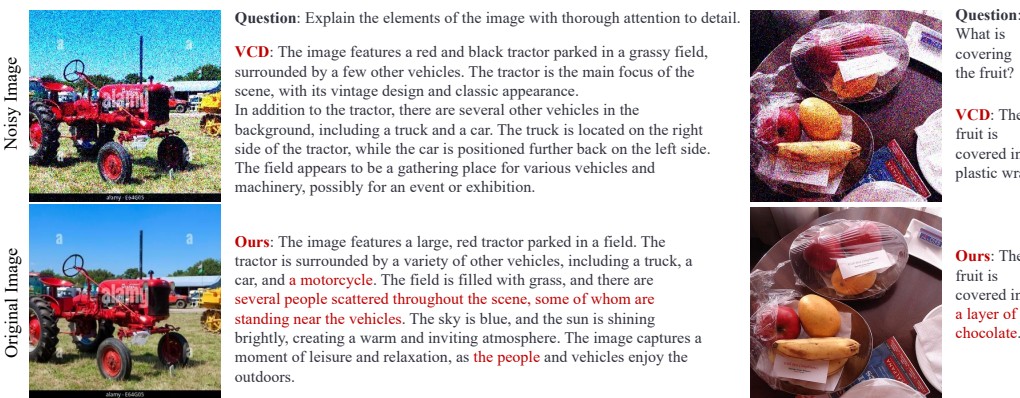

Figure 4: Model response generated by using negative image embeddings as inputs for positive image embeddings. For "VCD", we utilize noisy images as image inputs and for "Ours", we utilize negative image projection to project image inputs.

## 6 Conclusion

We introduce a novel method to mitigate hallucinations in MLLMs by decoupling the learning of positive and negative outputs through positive and negative image projections. This approach dynamically models authentic hallucination patterns, effectively suppressing contradictions without compromising general reasoning capabilities. Unlike training-based methods (*e.g.*, DPO) which suffer from the likelihood displacement issue, or training-free methods (*e.g.*, VCD) which rely on static perturbations, DCD optimizes vision-aware negative image features in contrastive decoding. This enables competitive hallucination reduction while maintaining performance in open-ended tasks. Our experiments demonstrate that focusing on negative (hallucinatory) samples significantly enhances the model's discriminative awareness, complementing the knowledge gained from supervised fine-tuning. This work advances the deployment of trustworthy MLLMs in high-stakes scenarios by striking a balance between accuracy and creativity.

## Acknowledgment

This work was supported by the National Natural Science Foundation of China Young Scholar Fund (62402408) and the Hong Kong SAR RGC Early Career Scheme (26208924).

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

# Appendix

## A    Social Impacts

Our work addresses the critical challenge of hallucination in multimodal large language models (MLLMs), with profound implications for the safe deployment of AI systems across socially sensitive domains. By mitigating factual contradictions and visual misrepresentations, our method enhances model reliability in high-stakes applications such as medical diagnostics and autonomous driving. Beyond safety-critical scenarios, our approach fosters trust in AI-assisted decision-making tools for education, legal documentation, and content moderation by ensuring outputs align with observable evidence. This reliability is particularly crucial for combating misinformation in an era of AI-generated content proliferation. Additionally, our findings on the importance of negative sample learning offer insights for developing more efficient alignment frameworks, potentially democratizing access to robust MLLMs for resource-constrained institutions.

## B    Limitations

While our method achieves robust hallucination mitigation, two key limitations warrant consideration. First, the contrastive decoding framework inherently doubles computational overhead during inference due to parallel processing of original and negative-projected image features. Second, our negative image projection relies on the quality and diversity of hallucination patterns in preference datasets. While current datasets predominantly cover common object hallucinations (*e.g.*, spurious object mentions), they may underrepresent complex multimodal hallucinations involving spatial reasoning or causal relationships. Future work could explore adaptive weighting mechanisms to handle such edge cases.

## C    Additional Experiments

**Results on Qwen 2.5 VL 3B**. As shown in Table A1 and Table A2, on a lightweight yet strong backbone, DCD preserves—or slightly improves—general capability while consistently suppressing hallucinations. Averaged over settings, our method surpasses DPO in three of four preference regimes, with "Pos.&Neg." typically the most stable variant. Overall, these appendix results support DCD's supervision- and model-agnostic transferability: learned negative embeddings—especially with joint positive-negative training—capture authentic hallucination patterns and extend cleanly to compact backbones, pointing to straightforward scaling on Qwen variants and broader preference signals.

## D    Theoretical Foundation

**1. Key Proposition: A Sufficient Condition for Eliminating** *Likelihood Displacement*

*From Pairwise to Decoupled Optimization.* DPO maximizes a *paired* log gap:

$$\mathcal{L}_{\text{DPO}}(\theta) = -\mathbb{E}_{(x,v,y^+,y^-)}\big[\log \sigma\big(\beta\left[\log \pi_\theta(y^+ \mid x,v) - \log \pi_\theta(y^- \mid x,v)\right]\big)\big],$$

which guarantees the gap widens but is **agnostic** to the absolute values of $\log \pi_\theta(y^+ \mid x,v)$ and $\log \pi_\theta(y^- \mid x,v)$. As a result, both likelihoods can **drift downward**—the *likelihood displacement* effect that degrades general reasoning (see Figure 1(a) **DPO** in the paper).

*Our Decoupled Objective.* Instead, we minimize two *independent* cross-entropies:

$$\min_\psi \ \mathbb{E}_{(x,v,y^+)}\big[-\log \pi_\theta\big(y^+ \mid x, g_\psi(v)\big)\big], \qquad \min_\phi \ \mathbb{E}_{(x,v,y^-)}\big[-\log \pi_\theta\big(y^- \mid x, g_\phi(v)\big)\big],$$

and combine the results **only at inference time**:

$$\widehat{\text{logit}} \ = \ (1+\alpha)\, \text{logit}_\psi \ - \ \alpha\, \text{logit}_\phi \,.$$

As shown in Figure 1(a) **Ours**, this ensures that $\log \pi_\theta(y^+ \mid x,v)$ increases while $\log \pi_\theta(y^- \mid x,v)$ is suppressed. This construction provides a **lower-bound guarantee on general reasoning performance**, consistent with our empirical results (Table 1, SEED-Bench).

## 2. Necessity and Sufficiency of the Negative Projector

*Theoretical View.* We interpret the negative image projection $g_\phi$ as learning an *adversarial negative distribution* $q_\phi(v)$ in the vision-feature space $\mathcal{V}$, which **maximizes**

$$\mathrm{KL}\Big(p(y \mid x, v) \,\big\|\, p(y \mid x, g_\phi(v))\Big),$$

equivalent to maximizing the InfoNCE lower bound. Thus, learning only the negative projector implicitly provides a *gradient-shaped penalty* during inference, which explains why our **Neg-only** training consistently reduces hallucination.

*Empirical Evidence.* As shown in Table 1, the **Neg-only** variant *nearly matches* **Pos + Neg** on hallucination benchmarks, and *clearly outperforms* **Pos-only**. To our knowledge, ours is the first work to show that negative-only learning can suffice.

| | General Performance | | | | Hallucination | | | | Average* |
|---|---|---|---|---|---|---|---|---|---|
| | SEED | MathVista† | MMStar | MMMU | MM-Vet† | MMHal† Score | MMHal† Rate ↓ | Hallusion† | |
| Qwen 2.5 VL 3B | 66.67 | 61.0 | 53.27 | 44.4 | 52.5 | 2.01 | 0.69 | 50.68 | 54.75 |
| + VCD [15] | 65.15 | 64.1 | 52.53 | 44.9 | 52.5 | 1.96 | 0.71 | 49.21 | 54.73 |
| *Fine-tuned on RLHF-V [16]* | | | | | | | | | |
| DPO [14] | 66.61 | 60.9 | 53.53 | 44.8 | 51.3 | 1.83 | 0.73 | 50.81 | 54.66 |
| SimPO [75] | 66.89 | 61.2 | 53.07 | 44.9 | 50.1 | 2.07 | 0.68 | 49.63 | 54.30 |
| Ours (Neg. Only) | $65.00_{-1.89}$ | $62.4_{+1.20}$ | $53.53_{+0.00}$ | $44.3_{-0.60}$ | $53.2_{+1.90}$ | $1.98_{-0.09}$ | $0.70_{+0.02}$ | $51.94_{+1.13}$ | $55.06_{+0.40}$ |
| Ours (Pos. & Neg.) | $66.30_{-0.59}$ | $62.7_{+1.50}$ | $54.80_{+1.27}$ | $44.4_{-0.50}$ | $54.5_{+3.20}$ | $2.43_{+0.36}$ | $0.57_{-0.11}$ | $52.37_{+1.56}$ | $55.85_{+1.19}$ |
| *Fine-tuned on BPO [18]* | | | | | | | | | |
| DPO [14] | 66.96 | 63.8 | 53.13 | 45.3 | 52.4 | 1.84 | 0.69 | 52.66 | 55.71 |
| SimPO [75] | 66.92 | 63.2 | 53.67 | 45.1 | 51.5 | 2.35 | 0.55 | 53.68 | 55.68 |
| Ours (Neg. Only) | $67.11_{+0.15}$ | $60.2_{-3.60}$ | $52.93_{-0.74}$ | $46.6_{+1.30}$ | $53.8_{+1.40}$ | $1.97_{-0.38}$ | $0.68_{+0.13}$ | $53.31_{-0.37}$ | $55.66_{-0.05}$ |
| Ours (Pos. & Neg.) | $67.12_{+0.16}$ | $63.2_{-0.60}$ | $53.07_{-0.60}$ | $46.9_{+1.60}$ | $53.2_{+0.80}$ | $2.55_{+0.20}$ | $0.52_{-0.03}$ | $53.26_{-0.42}$ | $56.13_{+0.42}$ |
| *Fine-tuned on RLAIF-V [17]* | | | | | | | | | |
| DPO [14] | 66.84 | 60.8 | 53.13 | 44.6 | 52.9 | 1.58 | 0.76 | 50.16 | 54.74 |
| SimPO [75] | 66.32 | 60.2 | 52.67 | 45.1 | 45.6 | 2.55 | 0.59 | 46.37 | 52.71 |
| Ours (Neg. Only) | $64.70_{-2.14}$ | $61.1_{+0.30}$ | $52.87_{-0.26}$ | $46.7_{+1.60}$ | $53.0_{+0.10}$ | $1.99_{-0.41}$ | $0.71_{-0.05}$ | $52.47_{+2.31}$ | $55.14_{+0.40}$ |
| Ours (Pos. & Neg.) | $65.61_{-1.23}$ | $60.9_{+0.10}$ | $50.27_{-2.86}$ | $45.8_{+0.70}$ | $53.5_{+0.60}$ | $2.30_{+0.72}$ | $0.61_{-0.15}$ | $52.68_{+2.52}$ | $54.79_{+0.06}$ |
| *Fine-tuned on VLFeedback [19]* | | | | | | | | | |
| DPO [14] | 66.74 | 60.5 | 52.33 | 44.6 | 53.1 | 2.14 | 0.66 | 48.27 | 54.26 |
| SimPO [75] | 66.96 | 62.3 | 52.87 | 44.6 | 51.9 | 2.33 | 0.68 | 48.05 | 54.45 |
| Ours (Neg. Only) | $65.80_{-1.16}$ | $61.2_{-1.10}$ | $53.13_{+0.26}$ | $45.2_{+0.60}$ | $53.0_{-0.10}$ | $2.21_{-0.12}$ | $0.66_{-0.00}$ | $51.79_{+3.52}$ | $55.02_{+0.57}$ |
| Ours (Pos. & Neg.) | $66.25_{-0.71}$ | $61.0_{-1.30}$ | $53.60_{+0.73}$ | $45.6_{+1.00}$ | $54.7_{+1.60}$ | $3.15_{+0.82}$ | $0.46_{-0.20}$ | $51.69_{+3.42}$ | $55.47_{+1.03}$ |

Table A1: Performance comparison on general and hallucination benchmarks for **Qwen 2.5 VL 3B**. 'Neg. Only' means only trained on negative samples of preference datasets, 'Pos. & Neg.' is trained with both positive and negative samples, ↓ indicates lower is better, and * denotes that the MMHal values are *not* counted in the Average score. † For benchmarks requiring GPT evaluation, we follow the same setting as the main table (e.g., GPT-4o 24-05-13).

| | Random | | Popular | | Adversarial | |
|---|---|---|---|---|---|---|
| | Acc | F1 | Acc | F1 | Acc | F1 |
| Woodpecker[76] | 85.51 | 86.67 | 83.51 | 84.33 | 82.35 | 83.00 |
| Qwen 2.5 VL 3B | 88.90 | 87.67 | 87.87 | 86.68 | 86.67 | 85.55 |
| + VCD [15] | 88.93 | 87.79 | 87.33 | 86.27 | 85.80 | 84.85 |
| *Fine-tuned on RLHF-V [16]* | | | | | | |
| DPO [14] | 88.83 | 87.58 | 87.83 | 86.61 | 86.60 | 85.46 |
| SimPO [75] | 88.07 | 86.59 | 87.27 | 85.82 | 86.23 | 84.84 |
| Ours (Neg. Only) | **89.57** | **88.54** | **88.30** | **87.32** | 86.80 | **85.92** |
| Ours (Pos. & Neg.) | 89.30 | 88.18 | 87.93 | 86.87 | **86.87** | 85.87 |
| *Fine-tuned on BPO [18]* | | | | | | |
| DPO [14] | 89.40 | 88.29 | 88.20 | 87.13 | **86.87** | 85.89 |
| SimPO [75] | 88.17 | 86.72 | 87.23 | 85.82 | 86.07 | 84.72 |
| Ours (Neg. Only) | **91.37** | **90.77** | 89.07 | 88.58 | 86.76 | **86.51** |
| Ours (Pos. & Neg.) | 90.90 | 90.20 | **89.33** | **88.70** | 86.63 | 86.23 |
| *Fine-tuned on RLAIF-V [17]* | | | | | | |
| DPO [14] | 89.23 | 88.08 | 88.10 | 86.99 | 86.67 | 85.64 |
| SimPO [75] | 89.57 | 88.46 | 88.40 | 87.33 | **86.90** | 85.93 |
| Ours (Neg. Only) | **91.37** | **90.80** | 88.40 | **88.02** | 86.07 | **85.95** |
| Ours (Pos. & Neg.) | 90.30 | 89.56 | **88.53** | 87.89 | 85.80 | 85.41 |
| *Fine-tuned on VLFeedback [19]* | | | | | | |
| DPO [14] | 87.77 | 86.21 | 87.07 | 85.53 | 86.00 | 84.52 |
| SimPO [75] | 87.03 | 85.20 | 86.40 | 84.59 | 85.50 | 83.74 |
| Ours (Neg. Only) | **90.03** | **89.08** | **88.47** | **87.58** | **87.07** | **86.28** |
| Ours (Pos. & Neg.) | 89.27 | 88.17 | 88.00 | 86.96 | 85.90 | 85.02 |

Table A2: Performance comparison on POPE [31] with Qwen 2.5 VL 3B. "Neg. Only" uses only negative samples from preference datasets; "Pos. & Neg." uses both positive and negative samples.

