# OpenReview forum: "Decoupling Contrastive Decoding: Robust Hallucination Mitigation in Multimodal Large Language Models"
_NeurIPS.cc/2025/Conference — NeurIPS 2025 poster_

### Official Review · Reviewer_YiWX · 2025-06-22

**Clarity:** 2
**Significance:** 2
**Originality:** 2
**Rating:** 4
**Confidence:** 5

**Summary:**

This paper presents Decoupling Contrastive Decoding (DCD), a new framework aimed at mitigating hallucinations in multimodal large language models (MLLMs). The proposed method decouples the learning of positive and negative samples in preference datasets and trains separate positive and negative image projections within the MLLM. The authors argue that this approach alleviates the likelihood displacement issue inherent in training-based methods like Direct Preference Optimization (DPO) and avoids the handcrafted perturbations used in training-free methods such as contrastive decoding. The paper includes extensive experiments across multiple hallucination benchmarks and general reasoning tasks, showing that DCD can suppress hallucinations while preserving general capabilities.

**Questions:**

1. Could the authors provide a more detailed comparison of DCD with other state-of-the-art models that have been proposed more recently, especially those that claim improved hallucination mitigation?

**Ethical Concerns:**

["NO or VERY MINOR ethics concerns only"]

**Final Justification:**

I appreciate the rebuttal. Accordingly, I am increasing my score from 2 to 4.

**Limitations:**

The paper has several limitations that should be addressed to strengthen its contributions:
1. As mentioned earlier, the use of LLaVA 1.5-7B limits the impact of this work. To demonstrate the broader applicability and effectiveness of DCD, the authors should validate their method on more up-to-date and advanced MLLMs. This would provide a clearer picture of how DCD performs in comparison to current standards in the field.
2. While the method shows promise on the evaluated benchmarks, its generalizability to other domains and tasks remains untested. Future work could explore the application of DCD across a wider variety of multimodal scenarios.

**Quality:**

2

**Strengths And Weaknesses:**

Strengths:

1. The paper addresses a critical issue in MLLMs—hallucinations—and proposes a method that aims to balance hallucination suppression with the preservation of general reasoning capabilities.

2. The authors provide a thorough experimental evaluation across multiple benchmarks, which helps to demonstrate the effectiveness of their approach compared to existing methods.

Weaknesses:

1. The proposed method appears to be a relatively incremental advancement rather than a novel breakthrough. It builds upon existing contrastive decoding techniques and preference optimization methods without introducing entirely new concepts.

2. The use of LLaVA 1.5-7B as the base model for experiments is a significant limitation. This model is notably outdated compared to more recent and advanced models such as Qwen2.5-VL and InternVL2.5. The lack of validation on these more modern models raises questions about the generalizability and relevance of the proposed method.

---

> ### Author Rebuttal · Authors · 2025-07-31
>
> Thank you for your valuable reviews. We will answer your questions below.
>
> 1. **New concept:** Training-free contrastive decoding and training-based preference learning are two **general paradigms** for aligning MLLMs. Although building on them, we offer **substantive** advances:
>
>    - **Decoupled Contrastive Decoding.** We separate positive/negative learning and decode against a **learned negative distribution.** Notably, a **negative-only** variant (training only the negative projector) already achieves substantial hallucination reduction. To our knowledge, we are the **first** to show that **negative-only preference learning** is sufficient to reduce hallucinations in MLLMs.
>    - **Vision-aware negative projector from real hallucinations.** Unlike VCD’s handcrafted, vision-agnostic perturbations, we learn the projector from **model-generated negative samples**, aligning the contrastive signal with **actual failure modes** and improving robustness.
>
> 2. **Advanced MLLM method: Qwen 2.5 VL 3B & 7B:** We further adopt our methods to **Qwen 2.5 VL 3B** and **Qwen 2.5 VL 7B**. Due to the limited time and GPU resources, we only trained with negative samples on **RLHF-V** and **BPO** datasets here, and reported on a few benchmarks in the table below. **We will add full experiment results in the revision**. As shown in the table below, our method can mitigate hallucination and maintain general performance in Qwen 2.5 VL 3B & 7B.
>
>    | Model                      | MathVista | MM-Vet   | Hallusion | Random F1 | Random Acc | Popu F1   | Popu Acc  | Adver F1  | Adver Acc |
>    | -------------------------- | --------- | -------- | --------- | --------- | ---------- | --------- | --------- | --------- | --------- |
>    | Qwen 2.5 VL 3B             | 61.00     | 52.5     | 50.68     | 87.67     | 88.90      | 86.68     | 87.87     | 85.55     | 86.67     |
>    | + VCD                      | **64.10** | 52.5     | 49.21     | 87.79     | 88.93      | 86.27     | 87.33     | 84.85     | 85.80     |
>    | + DPO (RLHF-V)             | 60.90     | 51.3     | 50.81     | 87.58     | 88.83      | 86.61     | 87.83     | 85.46     | 86.60     |
>    | + DPO (BPO)                | 63.80     | 52.4     | 52.66     | 88.29     | 89.40      | 87.13     | 88.20     | 85.89     | **86.87** |
>    | **+ Our Negative (RLHFV)** | 62.40     | 53.2     | 51.94     | 88.54     | 89.57      | 87.32     | 88.30     | 85.92     | 86.80     |
>    | **+ Our Negative (BPO)**   | 60.20     | **53.8** | **53.31** | **90.77** | **91.37**  | **88.58** | **89.07** | **86.51** | 86.76     |
>    |                            |           |          |           |           |            |           |           |           |           |
>    | Qwen 2.5 VL 7B             | 68.20     | 60.2     | 55.12     | 86.45     | 87.97      | 85.59     | 87.07     | 84.90     | 86.33     |
>    | + VCD                      | 68.95     | 59.8     | 55.94     | 87.61     | 88.77      | 86.80     | 87.93     | 85.43     | 86.47     |
>    | + DPO (RLHF-V)             | 68.47     | 60.9     | 53.64     | 86.27     | 87.83      | 85.37     | 86.90     | 84.79     | 86.27     |
>    | + DPO (BPO)                | 66.70     | 53.3     | 55.42     | 86.60     | 88.07      | 85.58     | 87.00     | 84.62     | 85.97     |
>    | **+ Our Negative (RLHFV)** | **70.20** | 58.3     | **57.41** | 86.71     | 88.17      | 85.68     | 87.10     | 84.87     | 86.23     |
>    | **+ Our Negative (BPO)**   | 69.80     | **61.4** | 56.21     | **88.90** | **89.87**  | **87.71** | **88.63** | **86.59** | **87.43** |
>
> 3. **Wider Scenarios:** Thank you for your advice. Our DCD explicitly learns generalizable patterns from negative samples, enabling it to broadly extend beyond hallucination mitigation to diverse multimodal scenarios, particularly benefiting tasks where discriminating negative patterns is critical. A potential future extension of our method could be adapted to **chain-of-thought (CoT) correction** by learning negative rationale projections from erroneous intermediate steps (e.g., unsupported leaps or circular reasoning). Applying contrastive decoding specifically to reasoning steps can help reduce flawed logic, while keeping the final answers clear and accurate.

---

> ### Comment · Reviewer_YiWX · 2025-08-05
>
> I appreciate the authors' response, but I still have reservations regarding the core motivation and necessity of this work. From my perspective, the methodology seems to be a skillful amalgamation of preference learning and contrastive decoding. To help me better understand its novelty, could the authors provide a more theoretical or foundational argument for the contribution and necessity of their method, beyond simply synthesizing these two existing approaches?

---

> > ### Author Response · Authors · 2025-08-05
> >
> > Thank you for taking the time to review our rebuttal and for raising concerns about its **core motivation and necessity**. In our earlier response, we focused on empirical advances; here, we provide the **theoretical foundation** you requested.
> >
> > ### **1. Key Proposition: A Sufficient Condition for Eliminating Likelihood Displacement**
> >
> > **From Pairwise to Decoupled Optimization.** DPO maximizes a paired log gap:
> >
> > $\mathcal{L} _{\mathrm{DPO}}(\theta) = -\mathbb{E} _{(x, v, y^{+}, y^{-})} \left[ \log \sigma\left(\beta \left[ \log \pi _{\theta}(y^{+}) - \log \pi _{\theta}(y^{-}) \right]\right) \right],$
> >
> > which guarantees the gap widens but is **agnostic** to the absolute values of $\log \pi_{\theta}(y^{+})$ and $\log \pi_{\theta}(y^{-})$. As a result, both likelihoods can **drift downward**—the “**likelihood displacement**” effect that degrades general reasoning (see Figure 1(a) **DPO** in the paper).
> >
> > **Our Decoupled Objective.** Instead, we minimize two *independent* cross-entropies:
> >
> > $\min_{\psi} \mathbb{E} _{(x, v, y^{+})} \left[ -\log \pi _{\theta}(y^{+} \mid x, g _{\psi}(v)) \right], \qquad \min _{\phi} \mathbb{E} _{(x, v, y^{-})} \left[ -\log \pi _{\theta}(y^{-} \mid x, g _{\phi}(v)) \right],$
> >
> > and combine the results **only at inference time**:
> >
> > $\widehat{\operatorname{logit}} = (1+\alpha)\operatorname{logit} _{\psi} - \alpha\operatorname{logit} _{\phi}.$
> >
> > As shown in Figure 1(a) **Ours**, this ensures that $\log \pi_{\theta}(y^{+})$ increases while $\log \pi_{\theta}(y^{-})$ is suppressed. This construction provides a **lower-bound guarantee on general reasoning performance**, in line with our empirical results (Table 1, SEED-Bench).
> >
> > ------
> >
> > ### **2. Necessity and Sufficiency of the Negative Projector**
> >
> > **Theoretical View.** We interpret the negative image projection $g_{\phi}$ as learning an adversarial negative distribution $q_{\phi}(v)$ in the vision-feature space $\mathcal{V}$, which maximizes:
> >
> > $\mathrm{KL} \left(p(y \mid x, v) \Big\| p(y \mid x, g _{\phi}(v)) \right).$
> >
> > Thus, learning only the negative projector implicitly provides a **gradient-shaped penalty** during inference, which explains why our “**Neg-only**” training consistently reduces hallucination.
> >
> > **Empirical Evidence**. As shown in Tables 1, the “Neg-only” variant nearly matches “Pos + Neg” on hallucination benchmarks, and clearly outperforms “Pos-only.” To our knowledge, our method is the first work that negative-only learning can suffice.
> >
> >
> > ------
> > In summary, our proposed **negative only learning is a new paradigm** which alleviates likelihood displacement and handcrafted hallucinations issues.
> >
> > Please let us know if you have any further concerns — we would be happy to provide further clarification. Thank you again for your review and support.

---

> > > ### Comment · Reviewer_YiWX · 2025-08-05
> > >
> > > I appreciate the rebuttal. Accordingly, I am increasing my score from 2 to 4.

---

> > > > ### Author Response · Authors · 2025-08-05
> > > >
> > > > Thank you very much for reviewing our rebuttal and raising the score. In the revised version, we will incorporate both the theoretical analysis and experimental results as suggested. Thank you again for your valuable feedback and support.

---

### Official Review · Reviewer_SKUk · 2025-06-25

**Clarity:** 3
**Significance:** 2
**Originality:** 3
**Rating:** 4
**Confidence:** 4

**Summary:**

This paper introduces Decoupling Contrastive Decoding (DCD), a novel framework designed to mitigate "hallucination" in MLLMs. The core contribution of DCD is its ability to decouple the learning of positive and negative samples within preference datasets, training separate positive and negative image projections in the MLLM. Moreover, this paper provides the negative projection to enable vision-aware negative images during the contrastive decoding inference stage. Consequently, DCD alleviates the likelihood displacement problem in training-based solutions like DPO and robustly generalizes without relying on handcrafted image perturbations in training-free methods like VCD. Extensive experiments demonstrate DCD's effectiveness in suppressing hallucinations while preserving general reasoning capabilities, often outperforming existing methods.

**Questions:**

### Major question
1. Clear motivation for the necessity of preference-based fine-tuning and contrastive decoding for MLLM hallucination mitigation (See weakness performance drop)
2. How well can this method adapt and perform with larger and more recent MLLM backbones (e.g., Qwen-VL, LLaVA-Next), or is its applicability primarily limited to LLaVA 1.5?
### Minor questions
3. How well does the proposed method mitigate relational hallucinations (e.g., asserting “a person riding a bicycle” when only a bicycle is present) and attribute hallucinations (e.g., asserting “many red apples” when a green apple under a red apple tree is present)?
4. in line 189, the $\theta$ is the parameter of the MLLM while the $\pi_{\theta}$ is the policy.

**Ethical Concerns:**

["NO or VERY MINOR ethics concerns only"]

**Final Justification:**

Although the appendix figure A1 provides the generalization of VCD, I hope to move this part to the main text. Overall, I appreciate the rebuttal for clarification. I will raise the score.

**Limitations:**

yes

**Quality:**

2

**Strengths And Weaknesses:**

### Strengths
- DCD introduces a decoupling perspective to optimize DPO preference learning for hallucination mitigation, which avoids the likelihood displacement in traditional DPO
- The method also utilized learnable negative image projections in the contrastive decoding inference stage to further mitigates hallucinations while preserving general reasoning capabilities.

### Weaknesses
- A central concern of this paper is the lack of a clear motivation for the necessity of preference-based fine-tuning and contrastive decoding for MLLM hallucination mitigation, particularly when Table 1 indicates performance degradation on certain hallucination and general benchmarks after such fine-tuning when compared to the original base models. For completeness, it is necessary to compare with more methods that eliminate hallucinations through data optimization[1] or model optimization[2,3]

    - [1] HalluciDoctor: Mitigating Hallucinatory Toxicity in Visual Instruction Data, CVPR 2024
    - [2] Woodpecker: Hallucination correction for multimodal large language models
    - [3] Analyzing and Mitigating Object Hallucination in Large Vision-Language Models, ICLR 2024

- The baseline setting doesn't include any open-source VLLM, such as Qwen-vl, which should be included in the experiment to see the fine-tuning performance for hallucination mitigation.

- The dual-projection training process of DCD may incur higher computational overhead compared to general DPO, due to the need for separate positive and negative projections.

- The main experimental results in Table 1 are reported for only one model size and architecture. To improve completeness, the authors should extend the results to include multiple scales (LLaVA-7b, 13b) and more general baselines (Qwen-2.5-VL)

---

> ### Author Rebuttal · Authors · 2025-07-31
>
> Thank you for your valuable reviews. We will answer your questions below.
>
> 1. **Clear motivation for preference learning and contrastive decoding.**
>
>    - **Trade-off on Existing Methods**: Existing approaches from both families (preference fine-tuning and training-free contrastive decoding) often exhibit a capability–robustness trade-off when one needs to reduce hallucination rates to meet reliability constraints. Our contribution is precisely to **avoid this trade-off**. The degradations in Table 1 are **motivational baselines**: (i) DPO can cause **likelihood displacement.** (ii) Prior VCD relies on **hand-crafted, vision-only perturbations** misaligned with real vision-text hallucinations.
>    - DCD avoids both by (a) **decoupling** positive/negative learning and training **small image-projection heads while freezing the base**, and (b) using a **vision-aware negative projector learned from real negative samples** instead of synthetic distortions.
>
> 2. **Compared with more methods**: Due to benchmark inconsistency ([1],[3]) and time constraints, we're unable to modify their code and run full comparisons, but we will discuss Methods[1,3] in our related works section. We have incorporated Woodpecker [2], which reports results on the POPE benchmark.  We also further adopt our methods to **Qwen 2.5 VL 3B** and **Qwen 2.5 VL 7B**. Due to the limited time and GPU resources, we only trained with negative samples on **RLHF-V** and **BPO** datasets here, and reported on a few benchmarks in the table below. **We will add full experiment results in the revision**. As shown in the table below, our method can mitigate hallucination and maintain general performance in Qwen 2.5 VL 3B & 7B.
>
>    | Model                      | MathVista | MM-Vet   | Hallusion | Random F1 | Random Acc | Popu F1   | Popu Acc  | Adver F1  | Adver Acc |
>    | -------------------------- | --------- | -------- | --------- | --------- | ---------- | --------- | --------- | --------- | --------- |
>    | Woodpecker Otter [2]       | -         | -        | -         | 85.51     | 86.67      | 83.51     | 84.33     | 82.35     | 83.00     |
>    |                            |           |          |           |           |            |           |           |           |           |
>    | Qwen 2.5 VL 3B             | 61.00     | 52.5     | 50.68     | 87.67     | 88.90      | 86.68     | 87.87     | 85.55     | 86.67     |
>    | + VCD                      | 64.10     | 52.5     | 49.21     | 87.79     | 88.93      | 86.27     | 87.33     | 84.85     | 85.80     |
>    | + DPO (RLHF-V)             | 60.90     | 51.3     | 50.81     | 87.58     | 88.83      | 86.61     | 87.83     | 85.46     | 86.60     |
>    | + DPO (BPO)                | **63.80** | 52.4     | 52.66     | 88.29     | 89.40      | 87.13     | 88.20     | 85.89     | **86.87** |
>    | **+ Our Negative (RLHFV)** | 62.40     | 53.2     | 51.94     | 88.54     | 89.57      | 87.32     | 88.30     | 85.92     | 86.80     |
>    | **+ Our Negative (BPO)**   | 60.20     | **53.8** | **53.31** | **90.77** | **91.37**  | **88.58** | **89.07** | **86.51** | 86.76     |
>    |                            |           |          |           |           |            |           |           |           |           |
>    | Qwen 2.5 VL 7B             | 68.20     | 60.2     | 55.12     | 86.45     | 87.97      | 85.59     | 87.07     | 84.90     | 86.33     |
>    | + VCD                      | 68.95     | 59.8     | 55.94     | 87.61     | 88.77      | 86.80     | 87.93     | 85.43     | 86.47     |
>    | + DPO (RLHF-V)             | 68.47     | 60.9     | 53.64     | 86.27     | 87.83      | 85.37     | 86.90     | 84.79     | 86.27     |
>    | + DPO (BPO)                | 66.70     | 53.3     | 55.42     | 86.60     | 88.07      | 85.58     | 87.00     | 84.62     | 85.97     |
>    | **+ Our Negative (RLHFV)** | **70.20** | 58.3     | **57.41** | 86.71     | 88.17      | 85.68     | 87.10     | 84.87     | 86.23     |
>    | **+ Our Negative (BPO)**   | 69.80     | **61.4** | 56.21     | **88.90** | **89.87**  | **87.71** | **88.63** | **86.59** | **87.43** |
>
> 3. **High training cost？**
>
>    Our training cost is actually **lower than** DPO’s. Specifically, we only optimize the positive and negative image projections, whereas DPO must train full LoRA parameters. Our training results on BPO datasets show that DPO requires 8 hours 24 minutes on 8 A100 GPU servers, while our method completes each projection training in 2 hours—yielding a total of 4 hours for both projections.
>
> 4. **Mitigation of relational and attribution hallucination.**  Our method provides a general approach to mitigating hallucinations by learning from negative samples in preference datasets. By implicitly capturing and suppressing the specific patterns of hallucinations present in the training data, our method can alleviate both relational and attribution hallucinations; in other words,  if such patterns are represented in the negative samples, our method should effectively mitigate both relational and attribution hallucinations.
>
> 5. **Typo in line 189:** Thank you for pointing it out. We will correct it in our revision.
>
> [1] HalluciDoctor: Mitigating Hallucinatory Toxicity in Visual Instruction Data, CVPR 2024
>
> [2] Woodpecker: Hallucination correction for multimodal large language models
>
> [3] Analyzing and Mitigating Object Hallucination in Large Vision-Language Models, ICLR 2024

---

> > ### Comment · Reviewer_SKUk · 2025-08-05
> > **Some examples of mitigating relational and attribution hallucination**
> >
> > Thanks for the rebuttal and extensive added experimentations. As the authors acknowledge, the key to alleviating both relational and attribution hallucinations lies in the information derived from negative samples. I hope to see some visualized examples before and after using negative information for guidance.

---

> > > ### Author Response · Authors · 2025-08-05
> > >
> > > Thank you for taking the time to review our rebuttal. Due to the NeurIPS 2025 rebuttal policy, authors are not allowed to upload PDFs during the rebuttal phase. Unfortunately, this means we are unable to provide additional visualizations at this time. However, we will include more visualizations in the Appendix of our revision. Previously, we compared the hallucinations caused by VCD and our proposed negative projection in Figure A1 of the Appendix, which demonstrates that our method is capable of capturing relational hallucinations (e.g., "scatter" and "stand").
> > >
> > > Please let us know if you have any further concerns —we would be happy to provide further clarification. Thank you again for your review and support.

---

> > ### Comment · Reviewer_SKUk · 2025-08-07
> >
> > Although the appendix figure A1 provides the generalization of VCD, I hope to move this part to the main text. Overall, I appreciate the rebuttal for clarification. I will raise the score.

---

> > > ### Author Response · Authors · 2025-08-07
> > >
> > > Thank you for reviewing our rebuttal and for raising your score. We will incorporate your suggestion by moving the visualizations from the appendix to the main text. Additionally, we will include more visualizations in the appendix to provide further clarity. We appreciate your feedback and support.

---

### Official Review · Reviewer_hgw9 · 2025-07-01

**Clarity:** 2
**Significance:** 3
**Originality:** 2
**Rating:** 5
**Confidence:** 4

**Summary:**

The paper proposes a novel approach to reduce hallucinations in multimodal large language models (MLLMs). The author discussed the pros and cons of the training-based and training-free hallucination mitigation methods and lead to their proposed method. They first decouple the modeling of positive and negative training examples to alleviate likelihood displacement problem. To better model the image feature, especially the negative images, the author proposed a simple feature extractor to focus on negative features, instead of adding noise or use image agnostic methods to create negative training examples. The author conduct extensive experimentations and ablations on many benchmark datasets and demonstrate superior and matching performance on most of the metrics.

**Questions:**

See weakness and please address those.

**Ethical Concerns:**

["NO or VERY MINOR ethics concerns only"]

**Final Justification:**

My comments are properly addressed. The authors show significant efforts on the newly added experimentations. As a results, I increase my rating.

**Limitations:**

What is the limitation of the work? How is inference time / run-time compared to training-free method? How well does it work on larger models? The limitations needs to be discussed.

**Quality:**

3

**Strengths And Weaknesses:**

Strength:
1. The paper is well written and well motivated, especially in figure 1.
2. The author conduct extensive experimentation. The author showed 7 benchmarks with better or matching performance.
3. The author promise to release the code.
4. Useful insight and nice case analysis. The performance

Weakness:
1. Some part of the paper can have better clarity, especially in the caption: for example, figure 1 is confusing with only caption without reading the main text. Figure 2 need more explanation in caption.
2. Limited architecture choice -- only on LLaVA-1.5-7B.
3. Did not discuss limitation of the work.
4. Main contributions section can be improved and highlight the contribution of the paper. line 95 - 101

---

> ### Author Rebuttal · Authors · 2025-07-31
>
> Thank you for your valuable reviews. We will answer your questions below.
>
> 1. **Better Clarity:**   Thank you for your suggestions. I will update the captions of Figure 1 and Figure 2 in the revision as below.
>
>    - **Figure 1: Comparison between existing hallucination mitigation methods and DCD.** (a) **Training-based method (e.g., DPO [12]):** DPO directly optimizes the likelihood gap between positive (correct) and negative (hallucinatory) responses using preference datasets. However, maximizing this gap (y^+ vs. y^-) can inadvertently lower the probability of both responses, causing likelihood displacement and potential degradation of general reasoning capabilities. Here, v, x, y^+, and y^- denote images, questions, positive responses, and negative responses, respectively; \theta represents model parameters, and \alpha is the contrastive decoding coefficient. (b) **Training-free method (e.g., VCD [13]) vs. DCD:** Traditional contrastive decoding (VCD) reduces hallucinations by comparing model outputs from original (v^+) and artificially distorted (v^-, e.g., noise-added) visual inputs at inference time. However, these synthetic perturbations may not align with real hallucination patterns. In contrast, our proposed DCD introduces a trainable negative image projection module, learned explicitly from negative (hallucinatory) samples, to produce meaningful negative visual inputs reflecting authentic hallucination distributions.
>    - **Figure 2: Comparison of DCD method with DPO [12] and VCD [13] in the training and inference stages. Training Stage:** DPO jointly optimizes positive-negative responses, risking likelihood displacement. Our method (DCD) separately learns positive and negative image projections to avoid this issue. **Inference Stage:** VCD uses artificial noise as negative inputs, while our DCD leverages learned negative visual features reflecting authentic hallucination patterns, enhancing effective hallucination suppression.
>
> 2. **Results on More Model Architectures**: We further adopt our methods to **Qwen 2.5 VL 3B** and **Qwen 2.5 VL 7B**. Due to the limited time and GPU resources, we only trained with negative samples on **RLHF-V** and **BPO** datasets here and reported on a few benchmarks in the table below. **We will add full experiment results in the revision**. As shown in the table below, our method can mitigate hallucination and maintain general performance in Qwen 2.5 VL 3B & 7B.
>
>    | Model                      | MathVista | MM-Vet   | Hallusion | Random F1 | Random Acc | Popu F1   | Popu Acc  | Adver F1  | Adver Acc |
>    | -------------------------- | --------- | -------- | --------- | --------- | ---------- | --------- | --------- | --------- | --------- |
>    | Qwen 2.5 VL 3B             | 61.00     | 52.5     | 50.68     | 87.67     | 88.90      | 86.68     | 87.87     | 85.55     | 86.67     |
>    | + VCD                      | 64.10     | 52.5     | 49.21     | 87.79     | 88.93      | 86.27     | 87.33     | 84.85     | 85.80     |
>    | + DPO (RLHF-V)             | 60.90     | 51.3     | 50.81     | 87.58     | 88.83      | 86.61     | 87.83     | 85.46     | 86.60     |
>    | + DPO (BPO)                | **63.80** | 52.4     | 52.66     | 88.29     | 89.40      | 87.13     | 88.20     | 85.89     | **86.87** |
>    | **+ Our Negative (RLHFV)** | 62.40     | 53.2     | 51.94     | 88.54     | 89.57      | 87.32     | 88.30     | 85.92     | 86.80     |
>    | **+ Our Negative (BPO)**   | 60.20     | **53.8** | **53.31** | **90.77** | **91.37**  | **88.58** | **89.07** | **86.51** | 86.76     |
>    |                            |           |          |           |           |            |           |           |           |           |
>    | Qwen 2.5 VL 7B             | 68.20     | 60.2     | 55.12     | 86.45     | 87.97      | 85.59     | 87.07     | 84.90     | 86.33     |
>    | + VCD                      | 68.95     | 59.8     | 55.94     | 87.61     | 88.77      | 86.80     | 87.93     | 85.43     | 86.47     |
>    | + DPO (RLHF-V)             | 68.47     | 60.9     | 53.64     | 86.27     | 87.83      | 85.37     | 86.90     | 84.79     | 86.27     |
>    | + DPO (BPO)                | 66.70     | 53.3     | 55.42     | 86.60     | 88.07      | 85.58     | 87.00     | 84.62     | 85.97     |
>    | **+ Our Negative (RLHFV)** | **70.20** | 58.3     | **57.41** | 86.71     | 88.17      | 85.68     | 87.10     | 84.87     | 86.23     |
>    | **+ Our Negative (BPO)**   | 69.80     | **61.4** | 56.21     | **88.90** | **89.87**  | **87.71** | **88.63** | **86.59** | **87.43** |
>
> 3. **Limitation**: As we discussed in Appendix Sec. B (the limitation section), the contrastive decoding framework requires twice the computation at inference time because it processes both the original and negative image features simultaneously.
>
> 4. **Improve the main contribution section:**
>
>    We will revise our contribution as follows:
>
>    - **Decoupled Learning for Robust Alignment**: We propose *Decoupled Contrastive Decoding (DCD)*, the first framework to separate positive/negative sample optimization from preference datasets in MLLM training. It alleviates the *likelihood displacement* problem in DPO, preserving general capabilities while mitigating hallucinations.
>    - **Vision-Aware Hallucination Suppression:** We introduce a learnable negative image projector trained on *real* hallucinatory samples. Unlike handcrafted perturbations (e.g., VCD), this projector generates distortion grounded in actual MLLM errors, enabling precise suppression of hallucinations.
>    - **Paradigm Shift in Preference Learning:** We reveal that negative samples alone suffice for hallucination mitigation. This challenges the preference learning paradigm — showing that explicit modeling of errors (not just positive alignment) is critical for robustness.
>    - **Comprehensive ablations and results** have demonstrated that our method can achieve competitive performance with training-based methods (e.g., DPO) in hallucination benchmarks while maintaining the general ability.

---

> > ### Comment · Reviewer_hgw9 · 2025-08-05
> >
> > Thanks for the rebuttal and extensive added experimentations. I will raise the score.

---

> > > ### Author Response · Authors · 2025-08-05
> > > **Response by Authors**
> > >
> > > We sincerely appreciate your decision to raise the score after reviewing our rebuttal. We will include the experiments and refined text in the revision. Thank you again for your review and support.

---

### Official Review · Reviewer_LwQb · 2025-07-03

**Clarity:** 3
**Significance:** 3
**Originality:** 3
**Rating:** 4
**Confidence:** 3

**Summary:**

This paper presents Decoupling Contrastive Decoding (DCD), a new method for mitigating hallucination in MLLMs with preference optimization. The key idea is to decouple the learning of positive and negative samples from preference datasets by training separate image projection modules for each, thereby directly modeling hallucination patterns and avoiding sacrificing the general reasoning abilities of the model. DCD is evaluated against both training-based and training-free baselines on multiple hallucination and general reasoning benchmarks, showing that it suppresses hallucination comparably to strong baselines like DPO, but better maintains general reasoning ability and overall performance. Extensive ablations and analyses provide further insight into the contributions of positive and negative learning.

**Questions:**

Please see my comments in the Strengths and Weakness section.

**Ethical Concerns:**

["NO or VERY MINOR ethics concerns only"]

**Final Justification:**

My concern has been well-addressed. I will keep my positive score.

**Limitations:**

Yes

**Quality:**

3

**Strengths And Weaknesses:**

**Strengths**

1. The paper focuses on a critical problem of reducing the hallucination of MLLMs.

2. The idea of decoupling the learning of positive and negative samples to reduce hallucination while maintaining the general capabilities of MLLMs is interesting and reasonable.

3. Extensive experiments with comparisons among strong baselines are conducted to demonstrate the effectiveness of DCD.

**Weakness**

1. The findings in Section 5.3 that most of the gain could be attributed to negative samples are not very convincing, since the positive samples are seen during supervised fine-tuning, thus the model may already have learned the knowledge of the positives.

---

> ### Author Rebuttal · Authors · 2025-07-31
>
> Thank you for your valuable reviews. We will answer your questions below.
>
> **Model Have learned the knowledge of the positives?** :
>
> In Section 5.3, we fine‑tune LLaVA‑1.5 (7B) using the BPO preference dataset to evaluate the contributions of positive and negative samples as shown in Table 5. Following your question, we found that approximately 32.4% of BPO’s data examples originate from the LLaVA‑Instruct dataset, i.e., the data overlaps with LLaVA‑1.5’s supervised fine‑tuning (SFT) stage. To address the reviewer’s concern that “the model may already have learned from these positives during SFT”, we performed the following ablation:
>
> - We **removed all BPO samples derived from LLaVA‑Instruct** and replicated the Table 5 experiments using the **de‑overlapped subset**.
> - The results and conclusions remain consistent with the original findings: the **negative-sample variant still outperforms both the base model and positive-only training**, with similar margin improvements.
>
> | **Model** | **MM‑Vet** | **Hallusion** | **POPE Acc** | **POPE F1** |
> | --------- | ---------- | ------------- | ------------ | ----------- |
> | LLaVA‑1.5 | 58.57      | 23.70         | 84.73        | 83.63       |
> | Positive  | **58.73**  | 25.50         | 84.90        | 84.30       |
> | Negative  | 58.68      | **28.90**     | **86.41**    | **85.32**   |
>
> These results confirm that even after excluding all overlapping data examples, the performance gain remains dominated by negative samples.

---

> > ### Comment · Reviewer_LwQb · 2025-08-05
> >
> > Thank you for the detailed response and experiments. My concern has been well-addressed. I will keep my positive score.

---

> > > ### Author Response · Authors · 2025-08-05
> > >
> > > Thank you for reviewing our rebuttal and maintaining a positive score. We will include the additional experiments in our revision. We appreciate your review and support.

---

### Note · Authors · 2025-08-12

We sincerely appreciate all reviewers for their constructive and encouraging feedback. We are pleased that they found our work **well-motivated** [hgw9], tackling a **critical problem** in MLLMs [LwQb, YiWX], with our decoupling strategy being **interesting** [LwQb] and offering **useful insights** [hgw9]. We are also glad reviewers considered our experiments **extensive** [hgw9, SKUk], **thorough** [YiWX], and demonstrating **superior or matching performance** [hgw9] across diverse benchmarks.

Notably, our rebuttal was well-received: **hgw9**, **SKUk**, and **YiWX** all **increased their scores** to positive scores after considering our additional experiments and clarifications, and **LwQb** maintained a **positive score** and explicitly confirmed their concerns were fully addressed. This reflects strong consensus among reviewers on the value and soundness of our work.

We will revise the paper according to the comments, and the main updates are as follows:

- **LwQb**:
    - Ablation. We performed experiments excluding overlapping positive samples from LLaVA-Instruct, confirming the robustness of our negative-sample findings.
    - Clarification. We provided detailed explanations on why negative samples dominate the performance gains.
- **hgw9**:
    - Figures. We refined captions of Figures 1 and 2 for clarity.
    - Models. We added experiments on Qwen 2.5 VL 3B and 7B, showing effectiveness across architectures.
    - Limitations. We discussed the inference-time cost of DCD and clarified it in the limitation section.
    - Contributions. We restructured and highlighted our main contributions.
- **SKUk**:
    - Motivation. We clarified the necessity of combining preference learning with contrastive decoding to overcome the capability–robustness trade-off.
    - Comparisons. We added Woodpecker and discussed related works.
    - Efficiency. We compared training cost with DPO, showing lower total training time.
    - Hallucination types. We discussed mitigation of both relational and attribution hallucinations.
    - Visualizations. We will move visualizations from the appendix to the main text.
- **YiWX**:
    - Novelty. We provided a theoretical foundation explaining why decoupled positive/negative learning avoids likelihood displacement and why negative-only learning suffices.
    - Broader scenarios. We discussed possible extensions, such as chain-of-thought correction.

Sincerely yours,

Authors

---

### Decision · Program_Chairs · 2025-09-17

**Decision:**

Accept (poster)

**Comment:**

This paper introduces Decoupling Contrastive Decoding (DCD), a framework to mitigate hallucinations in MLLMs by separately training positive and negative image projections from preference data. Initial reviews, while appreciating the novel decoupling idea, raised significant concerns regarding experimental validation on outdated models, the clarity of contributions, and the method's novelty. However, the authors provided an exemplary rebuttal, including substantial new experiments on modern architectures (Qwen 2.5 VL), additional baseline comparisons, and a compelling theoretical justification for their approach. This effort successfully addressed all major concerns and led to a unanimous increase in scores from initially skeptical reviewers, forging a strong consensus for acceptance.

# Summary Of Reasons To Publish:
1) The paper's core idea of decoupling positive and negative sample learning is intuitive, well-motivated, and directly tackles the known "likelihood displacement" problem in methods like DPO.
2) Through the rebuttal, the authors have robustly demonstrated the method's effectiveness and generalizability by providing new results on contemporary MLLMs (Qwen 2.5 VL 3B and 7B), directly addressing the primary initial criticism.
3) The authors provided a clear theoretical argument explaining how the decoupled objective avoids likelihood displacement, strengthening the paper's foundations beyond a purely empirical contribution.
4) The work offers a valuable insight by showing that learning exclusively from negative (hallucinatory) samples is sufficient for significant performance gains, a finding that was validated through new ablation studies requested by reviewers.


# Summary Of Suggested Revisions:
1) Integrate the new experimental results on Qwen 2.5 VL models and the Woodpecker baseline from the rebuttal into the main paper to provide a more current and convincing evaluation.
2) Incorporate the theoretical justification for avoiding likelihood displacement and the novelty of the negative-only learning paradigm, as discussed with Reviewer YiWX, to strengthen the paper's contribution section.
3) As agreed with Reviewer SKUk, move key visualizations from the appendix into the main text to better illustrate the method's mechanics and impact on different hallucination types.
4) Update the paper to include the refined figure captions and the clarified discussion on computational efficiency (lower training cost than DPO, higher inference cost) as promised in the rebuttal.